# Seeing Beyond the Crop: Using Language Priors for Out-of-Bounding Box Keypoint Prediction

**Bavesh Balaji**[1,2,3], **Jerrin Bright**[2,3], **Yuhao Chen**[2,3], **Sirisha Rambhatla**[1,3],
**John S. Zelek**[2,3], **David Anthony Clausi**[2,3],
[1]Critical ML Lab, [2]Vision and Image Processing Lab, [3]University of Waterloo,

## Abstract

Accurate estimation of human pose and the pose of interacting objects, like a hockey stick, is crucial for action recognition and performance analysis, particularly in sports. Existing methods capture the object along with the human in the bounding boxes, assuming all keypoints are visible within the bounding box. This necessitates larger bounding boxes to capture the object, introducing unnecessary visual features and hindering performance in real-world cluttered environments. We propose a simple image and text-based multimodal solution `TokenCLIPose` that addresses this limitation. Our approach focuses solely on human keypoints within the bounding box, treating objects as *unseen*. `TokenCLIPose` leverages the rich semantic representations endowed by language for inducing keypoint-specific context, even for occluded keypoints. We evaluate the performance of `TokenCLIPose` on a real-world ice hockey dataset, and demonstrate its generalizability through zero-shot transfer to a smaller Lacrosse dataset. Additionally, we showcase its flexibility on CrowdPose, a popular occlusion benchmark with keypoints within the bounding box. Our method significantly improves over state-of-the-art approaches on ice hockey, Lacrosse, and CrowdPose datasets, with gains of 4.36%, 2.35%, and 3.8%, respectively.

## 1 Introduction

The goal of 2D human pose estimation is to localize the human anatomical keypoints from an image, which is essential for scene understanding, action recognition [1, 2], and human-object interaction detection [3, 4]. This is particularly challenging in cluttered real-world scenarios due to occlusions and other non-idealities [5, 6]. With the emerging applications in Virtual Reality (VR), and Augmented Reality (AR), and real-time sports analysis [7], there is a fundamental need to understand how objects are manipulated via human-object interaction [8]. Often, in such scenarios, the objects that humans hold and interact with, which we define as *extensions*, can provide crucial information that aids in accurately estimating the human pose and the actions being performed [1].

Contemporary SOTA deep learning-based pose estimation methods predominantly follow a top-down approach: cropping each person in an image using bounding boxes before estimating their pose individually[9–16]. While using existing top-down pose estimators seems intuitive for joint prediction of the humans and their extensions, this approach suffers from limitations, yielding suboptimal results as shown in Fig. 1(a).

Our key observation is: capturing the extension in the bounding box expands the field-of-view and introduces unnecessary visual features, which can be confusing to the model. A simple yet powerful fix is to confine the bounding box to capture the human body, treating the extension's keypoints as *unseen*. This approach reduces background interference, as we do not explicitly capture the extension. However, it leads to the loss of important visual information about the extension, making them unseen.

38th Conference on Neural Information Processing Systems (NeurIPS 2024).

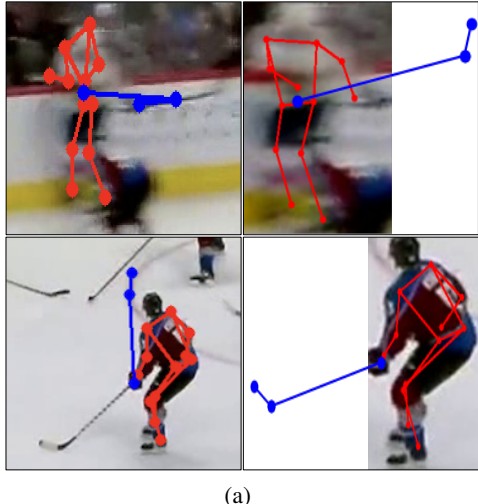 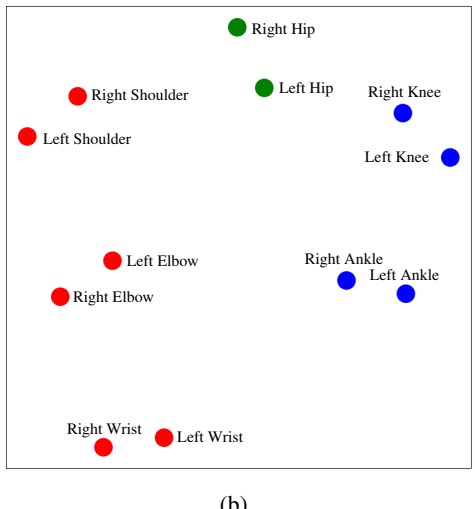

|       (a)       |       (b)       |

Figure 1: Difference between existing networks and our network. **(a)** Qualitative comparisons between HRNet (left) and our pose estimator (right). **(b)** t-SNE Visualization of keypoint-specific prompt embeddings. The different colours represent whether they are upper-body or lower-body joints, with red representing upper-body joints, blue representing lower-body joints and green representing the hip joints. From the figure, it is evident that while these embeddings maintain positional structure within the upper-body joints (the shoulders are placed above the elbows which are placed above the wrists, following the pose of a human standing normally) and the lower-body joints (knees placed above ankles), they fail to maintain positional structure between the upper-body and lower-body joints (elbows and wrists are placed below knees and ankles in the plot which does not follow the pose of a human being).

Here, a crucial question arises: *How can we effectively represent the spatial relationships of these unseen keypoints for accurate pose estimation?*

To answer this, we turn to other ways of informing the model about the unseen keypoints. Recent works have shown that using language to induce semantic context of keypoints can lead to effective feature representations [17–20]. Specifically, existing works [18] on human pose estimation align the image features with keypoint-specific text embeddings generated from Vision Language Models (VLMs) using a contrastive loss. However, these text embeddings primarily capture local details, neglecting the crucial global relationship between lower-body and upper-body joints; In Fig. 1(b) language models encode similar joints together, while losing the global structure. Hence, explicitly imposing the image features to be close to text embeddings could be suboptimal.

Based on the observations, we present a simple yet effective solution to significantly improve reliability under dynamic real-world scenarios by leveraging language to *see beyond the bounding box*. Specifically, We utilize these text embeddings as *priors* and initialize our learnable keypoint tokens (referred to as text tokens) using these text embeddings. By integrating the rich semantic representations of keypoint-specific text embeddings with image features, and employing a transformer to capture global dependencies, we extract superior fine-grained representations which significantly boosts the performance across the board.

We evaluate `TokenCLIPose`'s performance on three real-world datasets containing a lot of occlusions and noise: an ice hockey dataset, a Lacrosse Dataset and the CrowdPose dataset [6]. The ice hockey and Lacrosse datasets are first-of-their-kind datasets that we curated for predicting the pose of human extensions (the sticks). Furthermore, in order to demonstrate the flexibility of `TokenCLIPose` in predicting unseen keypoints that are present within the bounding box, we evaluate it on the CrowdPose dataset. `TokenCLIPose` outperforms existing top-down approaches by 4.36% and 3.8% on the ice hockey and CrowdPose datasets respectively. Furthermore, `TokenCLIPose` demonstrates superior zero-shot capabilities in predicting extension keypoints when tested on the Lacrosse dataset,

outperforming prior works by 2.35%. Our experiments highlight `TokenCLIPose`'s ability to reliably predict *unseen* keypoints.

## 2  Related Work

**2D Human Pose Estimation:** Top-down approaches in 2D pose estimation can be broadly classified into two categories: *Heatmap-based pose estimation* has been the *de-facto* standard approach since the introduction of stacked hourglass networks [14]. These methods represent discrete $(x, y)$ coordinates as continuous heatmaps where each pixel indicates the likelihood of a specific joint being located at that position. Most methods [9, 12, 14, 13] rely on powerful convolutional networks to extract high-level multi-scale feature maps. While most research focuses on network architectures, a few works investigate the coordinate representation and the heatmap encoding and decoding process [21, 15]. Recently, researchers have begun exploring transformer-based architectures for pose estimation [10, 22, 11, 23]. Xu *et al.* [10] adopt the original vision transformer [24] and build baselines for pose estimation, showcasing the efficacy of vision transformers. Methods including [22, 11] use CNNs as feature extractors, and utilize transformers to model the relationship between different scale features and the keypoint features respectively.

*Regression-based pose estimation*, in contrast to the dominance of heatmap-based approaches offers an alternative paradigm. Sun *et al.* [25] leverages a convolutional backbone to extract feature maps and then utilize an integral operation to directly regress keypoint coordinates. Li *et al.* [26] developed a novel pose regression model which aims at minimizing the distance between the predicted and underlying distribution. However, the global pooling operation used in [26] results in loss of spatial information that is crucial for reliable pose estimation. More recent works [27, 16, 28] include transformer-based architectures: [27, 16] use an encoder-decoder strategy and are based on DETR [29] and Deformable-DETR [30] respectively.

While heatmap-based methods have achieved high accuracy, they have some limitations: 1) These methods have a non-differentiable heatmap decoding method; 2) Heatmap representation often leads to quantization error; and 3) They are not designed to predict out-of-bounding-box keypoints. Therefore, we adopt the regression-based approach for directly estimating the coordinates of all joints, both inside and outside the bounding box.

**Occlusion-Aware Pose Estimation:** Several approaches tackle the problem of occlusions in crowded scenarios. Various methods tackle the problem by studying the relationships between multiple humans present in an image [31–33]. However, we do not compare with their works as we are interested in single-instance approaches. Park *et al.* [34] addresses the problem of *out-of-bounding box* keypoints by refining the bounding box before pose prediction, but this adds computational cost. Our approach, on the other hand, provides a parameter-free technique to induce spatial context for the *out-of-bounding box* keypoints by leveraging the knowledge of VLMs.

**Vision-Language Models:** Language supervision has been shown to improve feature representations in various vision tasks, such as image classification, semantic segmentation, and pose estimation. Radford *et al.* [35] proposed the contrastive pretraining paradigm CLIP that leverages contrastive learning to optimize a text and image encoder jointly. This work also showcased the significance of large-scale vision-language pretraining by demonstrating accurate zero-shot classification. Following CLIP, various works such as [36–38] focused on improving image classification using CLIP. Zhang *et al.* [39] transfer the 2D pre-trained knowledge to 3D domains, thereby improving zero-shot point-cloud recognition. Rao *et al.* [40] showcased the efficacy of CLIP pretraining on dense prediction tasks by converting the image-text matching problem to a pixel-text matching problem.

Recently, Tevet *et al.* [20] leverage text to inpaint missing poses in a sequence in a spatiotemporally consistent manner. Guo *et al.* [19] use pose-aware prompts to predict 3D hand meshes from images. Recent works on 2D pose estimation [18, 17] utilize joint-specific keypoints to learn richer representations. Particularly, they use a contrastive loss to align the image features extracted from a backbone to their keypoint-specific text embeddings. This, however, is suboptimal as the text embeddings do not capture global structure between human keypoints. Hence, aligning the image features to text embeddings might lead to loss of spatial information and inaccurate localization of joints. We improve this by using a transformer-based network to capture the global spatial dependencies between image and text features, thereby guiding our image features using text supervision rather than biasing them to the text prompts themselves.

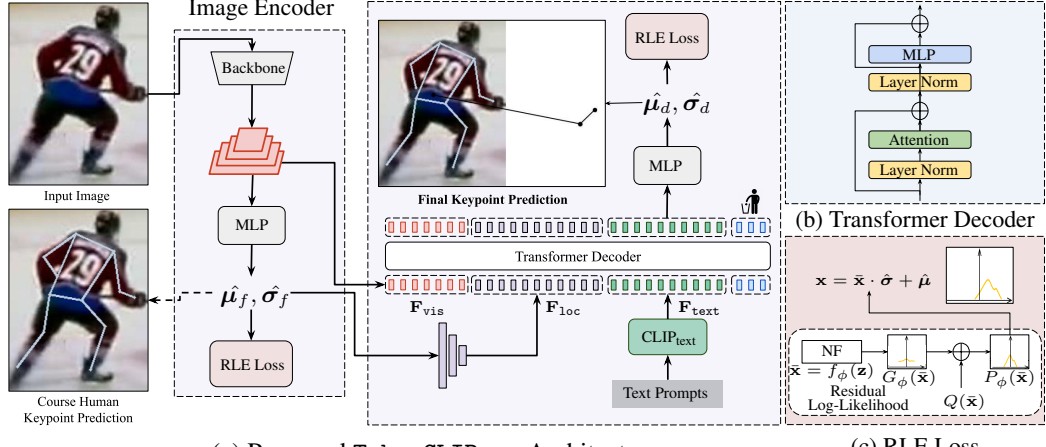

(a) Proposed `TokenCLIPose` Architecture  (b) Transformer Decoder

(c) RLE Loss

Figure 2: `TokenCLIPose` Architecture: We first incorporate an image encoder to extract multi-scale image features and coarse human keypoint locations $\hat{\boldsymbol{\mu}}_f$, and project them onto a joint multimodal embedding space obtaining image tokens $\mathbf{F}_{\texttt{vis}}$ and location tokens $\mathbf{F}_{\texttt{loc}}$ respectively. Then, we leverage a text-based keypoint encoder to extract keypoint-specific text tokens $\mathbf{F}_{\texttt{text}}$ from VLMs. These multimodal tokens are fed to a transformer decoder to capture spatial dependencies between them and predict all the 2D keypoints $\hat{\boldsymbol{\mu}}_d$. The coarse human keypoint predictions and the final keypoint predictions are supervised using the RLE loss.

# 3 Method

## 3.1 Problem Formulation

Given a cropped image $\mathbf{I} \in \mathbb{R}^{h \times w}$ of a human, generated from bounding boxes obtained through a detection network, we predict $K_{out}$ keypoints $\hat{\boldsymbol{\mu}}_d \in \mathbb{R}^{K_{out} \times 2}$ that represent the 2D poses of extensions and/or humans, along with the scale parameter $\hat{\boldsymbol{\sigma}}_d$ for each keypoint.

## 3.2 Network Architecture

We propose an encoder-decoder architecture to estimate 2D keypoints that are *not captured* in the bounding box. Firstly, an image encoder extracts multi-scale image features from the cropped image, which are then passed through a Multi-layer Perceptron (MLP) to generate coarse human keypoint proposals. These image features and locations are then projected onto the multimodal embedding space to form image and location tokens. The text-based keypoint encoder leverages text prompts to generate keypoint-specific text tokens. Finally, all these multimodal tokens are concatenated and passed through a transformer decoder to capture global relationships between these tokens and predict final 2D keypoints. The network is exemplified in Fig. 2.

**Image Encoder.** The cropped input image is initially passed through a pretrained CNN to extract multi-scale dense feature maps. These multi-scale feature maps are fused and projected onto a joint multimodal embedding space to form the image tokens $\mathbf{F}_{\texttt{vis}} \in \mathbb{R}^{N \times C_{emb}}$, where $N$ is the number of tokens and $C_{emb}$ is the joint multimodal space dimension. Furthermore, they are processed through a MLP to generate coarse human keypoint predictions $\hat{\boldsymbol{\mu}}_f \in \mathbb{R}^{K_h \times 2}$, and a scale parameter $\hat{\boldsymbol{\sigma}}_f \in \mathbb{R}^{K_h \times 1}$, which are optimized using the RLE process as detailed in Section 3.3.

**Text-based Keypoint Encoder.** Following image feature extraction, the keypoint encoder tackles the challenge of *unseen* keypoints in the image. We employ language-guided keypoint representations, where text prompts encode class-related information of each keypoint. By leveraging CLIP's pretrained text encoder, we generate the text tokens $\mathbf{F}_{\texttt{text}} \in \mathbb{R}^{K_{out} \times C_{emb}}$ which are then concatenated with the extracted image tokens. This process indirectly injects visual context of the missing keypoints into the model, even when it's not directly visible in the cropped input image. Additionally, we incorporate the coarse human keypoint predictions $\hat{\boldsymbol{\mu}}_f$ and convert them to location tokens by projecting

them onto the joint mulitmodal space $\mathbf{F}_{\texttt{loc}} \in \mathbb{R}^{K_h \times C_{emb}}$, and concatenate them with the image and text tokens. The final set of tokens fed to the decoder are denoted as $\mathcal{F} = \{\mathbf{F}_{\texttt{vis}}, \mathbf{F}_{\texttt{text}}, \mathbf{F}_{\texttt{loc}}\}$.

**Transformer Decoder.** To predict all the keypoints precisely, the relationships between these multimodal tokens ($\mathcal{F}$) must be captured accurately. Therefore, we leverage the transformer decoder to understand the inherent correlations between different text, image and location tokens. Instead of treating each modality separately and utilizing a cross-attention mechanism to understand the associations between them, we treat all tokens together as a homogeneous entity and employ the standard self-attention mechanism. This approach offers a simpler way to gain a holistic view of the relationships between all keypoints, locations, and image patches. The transformer layers are followed by MLPs to estimate the final keypoint predictions $\hat{\boldsymbol{\mu}}_d \in \mathbb{R}^{K_{out} \times 2}$ and the scale parameter $\hat{\boldsymbol{\sigma}}_d \in \mathbb{R}^{K_{out} \times 1}$. To reduce artifacts in feature maps and understand richer relationships, we employ additional register tokens as [41].

### 3.3 Loss Function

**Distribution Learning.** Following [26, 16], we formulate the regression task as a distribution learning problem and adopt Maximum Likelihood Estimation (MLE) to effectively predict the output coordinates. We use normalizing flows to estimate the deviation in predicted and ground truth keypoint distributions. In mathematical terms, given an input image $\mathcal{I}$, our network estimates a distribution $P_{\Theta,\Phi}(\mathbf{x}|\mathcal{I})$ representing the probability of the ground truth keypoint appearing at the location $\mathbf{x}$. Here, $\Theta$ and $\Phi$ denote the parameters of our pose network and the flow model $f_\phi$, respectively. The flow model $f_\phi$ acts as a refinement step, iteratively transforming a preset Gaussian distribution $\bar{\mathbf{z}} \sim \mathcal{N}(0, \text{Id})$ to capture the deviation of the predicted and ground truth distributions using the network's prediction ($\hat{\boldsymbol{\mu}}$ and $\hat{\boldsymbol{\sigma}}$).

Equation (1) depicts the mathematical formulation of the RLE loss, where $Q(\bar{\boldsymbol{\mu}}_g)$ is the preset Gaussian distribution, $G_\phi(\bar{\boldsymbol{\mu}}_g)$ is the learned distribution by the flow model, $\bar{\boldsymbol{\mu}}_g = (\boldsymbol{\mu}_g - \hat{\boldsymbol{\mu}})/\hat{\boldsymbol{\sigma}}$ represents the normalized difference between ground truth and predicted keypoints, and $s$ is a constant term.

$$\mathcal{L}_{RLE} = -\log Q(\bar{\boldsymbol{\mu}}_g) - \log G_\phi(\bar{\boldsymbol{\mu}}_g) - \log s + \log \hat{\boldsymbol{\sigma}} \tag{1}$$

Similar to [16], we supervise both the coarse predictions ($\hat{\boldsymbol{\mu}}_f, \hat{\boldsymbol{\sigma}}_f$) and the final predictions from the decoder ($\hat{\boldsymbol{\mu}}_d, \hat{\boldsymbol{\sigma}}_d$) using RLE. Hence, our final loss function is

$$\mathcal{L} = \mathcal{L}_{RLE}^f + \mathcal{L}_{RLE}^d \tag{2}$$

where,

$$\mathcal{L}_{RLE}^f = -\log P_{\Theta_f, \Phi_f}(\mathbf{x}|\mathcal{I})\Big|_{\mathbf{x}=\boldsymbol{\mu}_g} \quad \text{and} \quad \mathcal{L}_{RLE}^d = -\log P_{\Theta_d, \Phi_d}(\mathbf{x}|\mathcal{I})\Big|_{\mathbf{x}=\boldsymbol{\mu}_g}$$

Here, $\Theta_f$ and $\Phi_f$ denote the parameters of the backbone network and the flow model of the coarse keypoint predictions, and $\Theta_d$ and $\Phi_d$ denote the parameters of our decoder regression model and flow model of the final keypoint predictions.

## 4 Experiments

A critical challenge in evaluating pose networks for human extensions is the lack of publicly available datasets containing both the human and extension pose annotations. To address this limitation, we curate two new sports datasets, ice hockey and Lacrosse, featuring pose annotations for humans and their corresponding extensions. Experiments on these datasets demonstrate the performance of our model in estimating the pose of extension keypoints, which are not captured within the bounding box. We further evaluate the model's ability to predict unseen keypoints (due to occlusion/self-occlusion) within the bounding box using the benchmarked multi-person cluttered dataset, Crowdpose [6], validating the generalizability of our model.

For human pose estimation, the total keypoints ($K_{out}$) depend on extensions. Without extensions, $K_{out}$ equals the number of human keypoints ($K_h$). When extensions are present, $K_{out}$ increases to $K_h$ plus the number of extension keypoints ($K_e$).

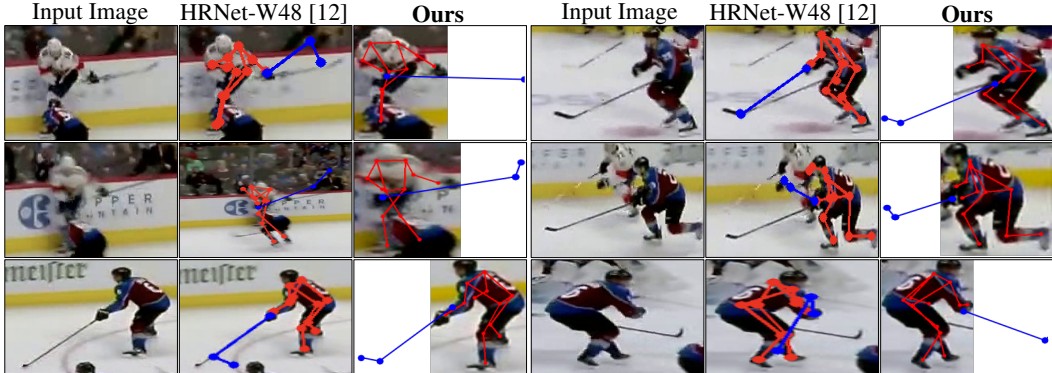

| Input Image | HRNet-W48 [12] | **Ours** | Input Image | HRNet-W48 [12] | **Ours** |

Figure 3: Qualitative Comparison of `TokenCLIPose` with HRNet-W48 on our ice hockey dataset.

## 4.1 Ice Hockey Dataset

**Dataset.** The ice hockey dataset, captured from real-world National Hockey League (NHL) videos, presents a challenging environment for pose estimation due to the inherent fast pace of the sport. Players' rapid movements often result in motion blur within frames, making it difficult to distinguish body parts. Furthermore, the nature of the game leads to frequent occlusions, especially when players obstruct each other's bodies. Adding to the complexity is the bulky equipment worn by hockey players, which can obscure keypoints. These combined challenges- motion blur, occlusion, and bulky equipment- make the ice hockey dataset a valuable resource for evaluating the robustness of the pose estimation task.

Our dataset consists of 10 video clips (30-45 seconds each, sampled at 30 fps) from various broadcast NHL videos. We utilize the CVAT tool to annotate the keypoints for multiple players and their hockey sticks in each frame. We followed the standard COCO format to annotate 17 keypoints for the human pose. Additionally, we annotate 3 keypoints for the hockey stick (butt end, heel, and toe). Finally, for each annotated frame, we create two versions of the input image: one cropped to include only the player (*standard bounding box*) and another incorporating both the player and their hockey stick (*extended bounding box*). In total, we generate $11.66K$ pose annotations, with $9.13K$ images from 9 clips used for training and $2.53K$ images from the tenth clip used for testing our model's performance.

**Evaluation Metric.** We evaluate different State-Of-The-Art (SOTA) pose estimation models on our dataset using the Percentage of Correct Keypoints with head-normalization (PCKh) metric. Due to our method's use of smaller bounding box compared to the other models, achieving the same level of accuracy would result in a higher PCKh threshold for our model. To ensure a fair and direct comparison across all models, we employ the same threshold for evaluation.

**Training.** All networks are trained for 200 epochs with a batch size of 64 on a single NVIDIA GeForce RTX 4090 GPU. We employ the Adam optimizer with a learning rate of $6 \times 10^{-4}$ for all CNN-based architectures while the transformer-based architectures are trained using the AdamW optimizer with an initial learning rate of $3 \times 10^{-4}$. The weight decay is set to $10^{-5}$ for all models. A stepLR scheduler was used to linearly reduce the learning rate from the initial value of $10^{-5}$. Consistent with [26], we utilize RealNVP [42] as the flow model within our model.

**Results.** The results, presented in Table 1, emphasize our method's *SOTA performance* over existing methods in robustly predicting the ice hockey player and the hockey stick keypoints, outperforming existing works by 4.36%. This is further validated by the qualitative comparisons in Fig. 5, where our model demonstrates the ability to predict semantic poses even in challenging scenarios with extreme motion blur and occlusion. These results support our hypothesis that *re-considering the extension pose estimation task as an unseen keypoint problem without explicitly capturing it in the bounding box* reduces background noise leading to robust poses.

## 4.2 Lacrosse Dataset

**Dataset.** In order to study the efficacy of `TokenCLIPose`'s generalization capabilities in predicting the pose of extensions, we curate another small-scale Lacrosse dataset. Similar to ice hockey, it is

Table 1: **Comparison with SOTA Methods** on our real-world ice hockey dataset (PCKh@0.5). **BoldFace** represents the best score. Underline represents the top score in existing works.

| Method | Backbone | Input Resolution | Body | Butt End | Stick Heel | Stick Toe | Mean |
|---|---|---|---|---|---|---|---|
| SimpleBaseline [13] | ResNet-50 | 256x192 | 93.59 | 69.57 | 57.19 | 52.76 | 68.83 |
| MSPN [9] | - | 256x192 | 93.61 | 70.30 | 59.21 | 55.69 | 69.70 |
| HR-Net [12] | HRNet-W48 | 256x192 | 94.90 | 71.48 | 60.29 | 55.36 | 70.44 |
| TokenPose-L/D24 [11] | HRNet-W48 | 256x192 | 95.13 | 70.96 | 60.93 | 56.27 | 70.82 |
| ViTPose [10] | ViT-B | 256x192 | 95.61 | 71.94 | 61.33 | 58.80 | 71.92 |
| TokenCLIPose | ResNet-50 | 256x192 | 95.81 | 74.86 | 65.79 | 65.08 | 74.92 |
| TokenCLIPose | MSPN | 256x192 | 97.17 | 75.41 | 66.70 | 66.34 | 75.53 |
| TokenCLIPose | HRNet-W48 | 256x192 | **97.37** | **75.94** | **67.82** | **66.15** | **76.28** |
| Improvement | - | - | 1.76% ↑ | 4.00% ↑ | 6.49% ↑ | 7.35% ↑ | 4.36% ↑ |

Table 2: **Zero-shot Comparison with SOTA Methods** on our real-world Lacrosse dataset (PCKh@0.5). **BoldFace** represents the best score. Underline represents the second-best score.

| Method | Backbone | Body | Butt End | Stick Heel | Mean |
|---|---|---|---|---|---|
| SimpleBaseline [13] | ResNet-50 | 94.73 | 67.28 | 53.99 | 72.00 |
| MSPN [9] | - | 95.84 | 70.68 | 57.40 | 74.64 |
| HR-Net [12] | HRNet-W48 | 95.92 | 71.35 | 58.41 | 75.22 |
| ViTPose [10] | ViT-B | 95.77 | 72.85 | 60.18 | 76.26 |
| TokenCLIPose | HRNet-W48 | **97.24** | **76.60** | **65.01** | **78.61** |
| Improvement | - | 1.47% ↑ | 3.75% ↑ | 4.83% ↑ | 2.35% ↑ |

characterized by motion blur and occlusions, but there are domain differences between the 2 datasets. This dataset consists of 300 pose annotations from one video sampled at 30 fps. We use the same 12 human keypoints used in the ice hockey dataset to denote the human's pose. However, we use only 2 keypoints to represent the Lacrosse stick's pose, as the blade of a Lacrosse stick is circular in nature. Hence, we disregard the 15th keypoint (stick toe) and use the other 14 keypoints to represent the pose of a Lacrosse player along with their stick.

**Zero-shot Results.** We evaluate TokenCLIPose's generalization capability by performing zero-shot transfer from our pretrained model on ice hockey to the Lacrosse dataset. Due to the difference in the shape of the head of a lacrosse and hockey stick, we predict the two keypoints corresponding to the shaft of a lacrosse stick. Furthermore, the text prompts for the two extension keypoints are also changed while keeping the model frozen. The results presented in Table 2 showcase that our model outperforms the established baselines by 2.35%, thereby demonstrating the efficacy of our proposed model for generalizable pose estimation.

### 4.3 CrowdPose Dataset

**Dataset.** The CrowdPose dataset is a large-scale benchmark dataset for human pose estimation, containing $12K$ images and $43.4K$ labeled people in their trainval set, and $8K$ images with $29K$ labeled people in the test set. Following [32, 18, 31], we use the trainval set for training and test set for evaluation.

**Evaluation Metric.** We adopt standard Average Precision (AP) as our evaluation metric on the CrowdPose dataset. AP is calculated based on Object Keypoint Similarity (OKS) denoted by $m_{OKS} \in \mathbb{R}$, which is defined as

$$m_{OKS} = \frac{\sum_i \exp(-\hat{d}_i^2/2s^2k_i^2)\sigma(v_i > 0)}{\sum_i \sigma(v_i > 0)}, \qquad (3)$$

where $\hat{d}_i$ is the Euclidean distance between the $i$-th predicted keypoint coordinate and the corresponding ground truth, $v_i$ is the visibility flag of the keypoint, $s$ is the object scale, and $k_i$ is a keypoint-specific constant.

**Training.** We employ the AdamW optimizer with an initial learning rate of $6 \times 10^{-4}$ and weight decay of 0.1. Following ViTPose [10], we apply linear warmup for the first 2000 iterations with a warmup factor of $10^{-3}$. Furthermore, we perform gradient clipping to prevent overfitting.

Table 3: **Comparison with SOTA Methods** on CrowdPose dataset. **BoldFace** represents the best score. Underline represents the second-best score.

| Method | Input Resolution | $AP$ | $AP_{50}$ | $AP_{75}$ | $AP_E$ | $AP_M$ | $AP_H$ |
|---|---|---|---|---|---|---|---|
| Mask-RCNN [43] | $256 \times 192$ | 57.2 | 83.5 | 60.3 | 69.4 | 57.9 | 45.8 |
| AlphaPose | $256 \times 192$ | 61.0 | 81.3 | 66.0 | 71.2 | 61.4 | 51.1 |
| SimpleBaseline [13] | $256 \times 192$ | 60.8 | 81.4 | 65.7 | 71.4 | 61.2 | 51.2 |
| CrowdPose [6] | $256 \times 192$ | 66.0 | 84.2 | 71.5 | 75.5 | 66.3 | 57.4 |
| Hourglass-104 [44] | $384 \times 288$ | 65.2 | 85.9 | 69.5 | - | - | - |
| KAPAO-L [45] | $384 \times 288$ | 68.9 | 89.4 | 75.6 | 76.6 | 69.9 | 59.5 |
| HRNet-W48 [12] | $384 \times 288$ | 69.3 | 89.7 | 75.6 | 77.7 | 70.6 | 57.8 |
| Transpose-H [22] | $384 \times 288$ | 71.8 | 91.5 | 77.8 | 79.5 | 72.9 | 62.2 |
| HRFormer-B [23] | $384 \times 288$ | 72.4 | 91.5 | 77.9 | 80.0 | 73.5 | 62.4 |
| TokenCLIPose | $384 \times 288$ | **76.2** | **93.9** | **82.4** | **83.3** | **77.4** | **66.1** |
| Improvement | - | **3.8%** ↑ | **2.4%** ↑ | **4.5%** ↑ | **3.3%** ↑ | **3.9%** ↑ | **3.7%** ↑ |

Table 5: **Effect of Attention Mechanisms** on the overall accuracy.

| Attention-Mechanism | Mean |
|---|---|
| Intention [46] | 75.14 |
| Self-attention | 76.28 |
| Cross-attention | **76.31** ↑ 0.03 |

Table 6: **Effect of Text Prompts** on the stick accuracy.

| Prompt type | Stick Accuracy | Mean |
|---|---|---|
| No text | 63.37 | 72.93 |
| Single Prompt | 67.41 | 75.24 |
| Prompt Ensemble | **69.97** ↑ 2.56% | **76.28** ↑ 1.04% |

**Results.** Table 3 presents a comparison between established pose estimation techniques and our method. As shown in the table, TokenCLIPose outperforms all the top-down approaches by 3.8% demonstrating its efficacy in predicting unseen keypoints robustly. We also depict the qualitative results of TokenCLIPose to depict its robustness to occlusion. It is notable that even in scenarios where we only see the side-view of humans, TokenCLIPPose estimates the pose reliably.

## 4.4 Ablation Studies

We conduct comprehensive ablations to study the effect of our design choices and verify the impact of each module on our proposed TokenCLIPose. For consistency, all the ablations are conducted on our ice hockey dataset unless specified otherwise.

**Gains from Each Modality.** The influence of each modality on the proposed model's performance is shown in Table 4. As evidenced by a performance improvement of 3.35%, the inclusion of text tokens plays a significant role in enhancing the pose estimation accuracy. On the other hand, we find that including the location tokens do not improve the performance significantly, only by a small margin of 0.45%.

Table 4: **Effect of Different Modalities** on the overall accuracy.

| Text tokens ($\mathbf{F}_{\texttt{text}}$) | Location tokens ($\mathbf{F}_{\texttt{loc}}$) | Image tokens ($\mathbf{F}_{\texttt{vis}}$) | Mean |
|---|---|---|---|
| ✗ | ✗ | ✓ | 72.48 |
| ✗ | ✓ | ✓ | 72.93 |
| ✓ | ✗ | ✓ | 75.72 |
| ✓ | ✓ | ✓ | **76.28** |

**Do we need to treat each modality heterogeneously?** We probe whether multimodal tokens need to be treated heterogeneously by testing out different attention mechanisms for our transformer decoder. As shown in Table 5, utilizing self-attention directly on all the tokens produces similar results to performing self-attention separately on image, location and text tokens, and then applying cross-attention. Therefore, it is not necessary to treat the tokens from each modality separately. We

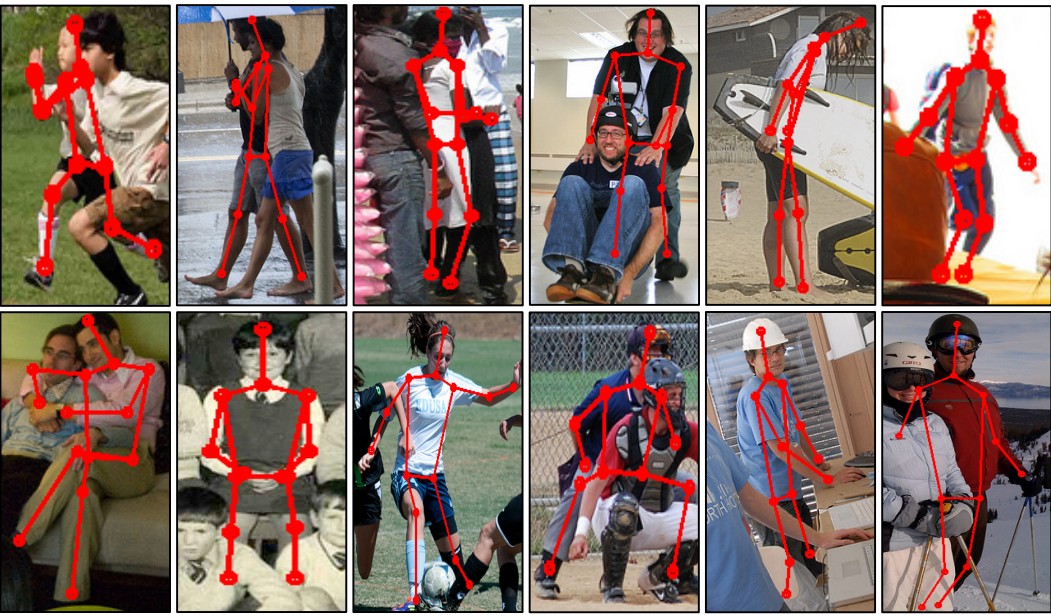

Figure 4: Qualitative Results of our `TokenCLIPose` on the CrowdPose dataset.

hypothesize that this could be due to all the tokens being projected to the same joint embedding space, thereby eliminating the need to process them differently.

**Impact of Text Prompts.** We investigate the influence of language on our model's efficacy on the ice hockey dataset by varying the degree and complexity of text prompts that we use. Starting from randomly initializing the text tokens instead of using CLIP embeddings, we study the effects of using single prompts and the prompt ensemble technique proposed in [35] for ImageNet classification. It is evident from Table 6 that incorporating text instead of randomly initializing text tokens results in a significant improvement of 2.31% in the overall accuracy. Furthermore, using prompt ensemble techniques improve the accuracy of the model by 1.04%, showcasing the importance of the quality of text prompts.

**Influence of Bounding Boxes.** We evaluate the influence of bounding box predictions on our model's performance by using ground truth bounding boxes obtained from ground truth human poses and the bounding boxes from the FasterRCNN object detector. The results are showcased in Table 7. The above table illustrates our model's robustness to the quality of bounding boxes, as using ground truth bounding boxes improves the model's performance by a small margin of 2.47%.

Table 7: **Impact of Bounding Boxes** on the overall accuracy.

| Bounding boxes | Mean |
| --- | --- |
| Faster-RCNN [47] | 76.28 |
| Ground Truth | **78.75** ↑ 2.47% |

## 5   Conclusion

In this work, we proposed `TokenCLIPose`, an innovative solution for robustly predicting human and *extension* poses. Instead of explicitly modeling extensions within the bounding box, we reformulated the *extension* pose estimation as an unseen keypoint prediction problem. We leverage the power of large pre-trained VLMs in augmenting the spatial information of unseen keypoints. We showcased that capturing relationships between multimodal tokens is more effective than aligning image features to text tokens. To evaluate the effectiveness of `TokenCLIPose` in predicting *extension* keypoints, we curated real-world datasets for ice hockey and Lacrosse. We significantly outperform existing top-down methods on these datasets (by 4.36% and 2.35%, respectively). Additionally, we achieve a 3.8%

improvement over prior top-down networks on the CrowdPose dataset. This shows `TokenCLIPose`'s flexibility to predict unseen keypoints within the bounding box as well. Future work will focus on curating and training more extensive and diverse datasets for human and extension pose estimation tasks.

## 6 Acknowledgement

This work was supported in part by Stathletes, Compute Canada, the Natural Sciences and Engineering Research Council of Canada and MITACS.

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

# A Appendix

In this appendix, we provide more qualitative results on our ice hockey and Lacrosse datasets. Furthermore, we provide a quantitative comparison with a multimodal network ([18]).

## A.1 Comparison with Multimodal Pose Estimators

To investigate the effectiveness of our technique of leveraging text with existing multimodal methods, we compare `TokenCLIPose`'s performance with the multimodal counterpart LAMP [18]. Though LAMP does not follow the top-down approach, it is the only network that uses language for 2D human pose estimation. Thus, we compare and highlight the results in Table 8. As shown in the table, it is evident that `TokenCLIPose` outperforms LAMP on the CrowdPose dataset by 4.8%. This corroborates our claim that *leveraging text as supervisory signals to guide image features provides better performance than aligning image features to the text tokens.*

Table 8: **Comparison with LAMP** on CrowdPose dataset

| Method | Input Resolution | $AP$ | $AP_{50}$ | $AP_{75}$ | $AP_E$ | $AP_M$ | $AP_H$ |
|---|---|---|---|---|---|---|---|
| LAMP [18] | $512 \times 512$ | 71.4 | 90.3 | 77.1 | 77.9 | 72.1 | 64.2 |
| `TokenCLIPose` | $384 \times 288$ | **76.2** | **93.9** | **82.4** | **83.3** | **77.4** | **66.1** |

## A.2 Qualitative Results

We showcase additional qualitative results on our ice hockey dataset visualizing the effectiveness of our approach. Furthermore, we also demonstrate the generalization capabilities of our model in transferring to a Lacrosse dataset. The impact of language is visible in our model's robustness to domain changes.

To further illustrate the effectiveness of our approach, we present additional qualitative results on the ice hockey dataset in Figure 5. Furthermore, we demonstrate the model's generalization capabilities by achieving strong performance on a Lacrosse dataset, highlighting the impact of language on the model's robustness to domain changes.

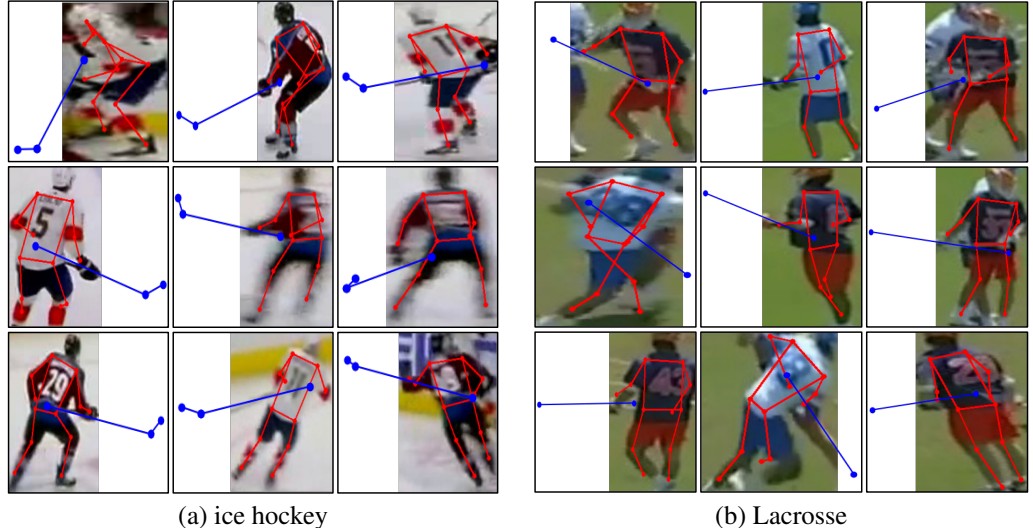

(a) ice hockey      (b) Lacrosse

Figure 5: Qualitative Results of `TokenCLIPose` for the extension pose estimation task on our ice hockey and Lacrosse datasets. We plot the pose of extensions outside the cropped image to understand how `TokenCLIPose` works.

## B  Broader Impact Statement

Our work provides a new way to reconsider the pose estimation task to predict out-of-bounding box keypoints. This improves our understanding of why joint hockey stick pose prediction along with player pose is difficult for existing top-down networks and presents one way to mitigate the issues. This understanding should enable the application of top-down solutions for multi-instance pose estimation which encounters similar issues. Furthermore, accurate stick pose prediction implies that hockey teams can make more data-driven decisions to improve their teams' performance without incurring additional overhead expenses. This also implies that the use of invasive technology such as infrared sensors can be avoided to a large extent to robustly analyze players and teams.

This can be leveraged in various domains where humans closely interact with objects, such as shoveling, where it could be used to study fatigue and biomechanics. Furthermore, as shown in our experiments on the CrowdPose dataset, `TokenCLIPose` also reliably predicts the pose of a human when there is high human-human interaction. Hence, this can be used for pose estimation in highly crowded scenarios, such as surveillance and monitoring people in crowds.

