# OpenReview forum: "Seeing Beyond the Crop: Using Language Priors for Out-of-Bounding Box Keypoint Prediction"
_NeurIPS.cc/2024/Conference — NeurIPS 2024 poster_

### Official Review · Reviewer_9aCe · 2024-06-26

**Soundness:** 3
**Presentation:** 2
**Contribution:** 3
**Rating:** 5
**Confidence:** 4

**Summary:**

This paper focuses on an interesting and inherent problem in top-down human pose estimation; that is out-of-box prediction in a top-down paradigm. The core of the solution is to utilize the semantic context by giving proper text prompts to CLIP. Specifically, a cropped image is first given to a pre-trained CNN model to extract initial features and obtain a coarse prediction. Then the visual features from CNN, the coarse prediction of the person, and the semantic information from CLIP are concatenated and input to a transformer decoder to obtain the final prediction. The method is capable of solving the occluded points in the crowded scene and breaking the limits from the bounding box in the top-down model.

**Strengths:**

1. This paper focuses on a very interesting topic that indeed requires a deeper investigation.

2. Having semantic information from text prompts is a quite novel and effective way of solving the occlusion and out-of-box points.

3. The paper is in general well-written and easy to follow.

**Weaknesses:**

1. Poor visualisation. Qualitative results are only compared with HRNet. More visualization should be shown to validate the effectiveness of the proposed method.

2. A few images are very ambiguous. Especially, the Figure 1. (b). There is no notation for x, y, and colors. The caption said it is the text embedding and such embedding loses the global structure. Why did it lose the global structure?

3. In terms of performance, there is indeed an improvement. However, how many of them are coming from the unseen points? i.e. Is the source of improvement from unseen points or a more accurate estimation of the seen points? After all, the whole pipeline is a coarse-to-fine prediction with additional text information. Therefore, the seen points should also benefit. If the major improvement does not come from the unseen points, then the major problem proposed by this paper still remains unsolved.

**Questions:**

1. One statement in the introduction that I do not agree with is "Contemporary deep learning-based pose estimation methods predominantly follow a top-down approach" between line 27 and line 28. Bottom-up and one-stage models are also important parts of human pose methods. In fact, bottom-up models have their own merits such as real-time inference and bounding-box independent predictions. I suggest revising this statement.

2. It would be better if the authors could make a further explanation on using normalizing flow to estimate the human poses. I think it is quite interesting as this method was published in 2021 and is quite rare in recent work. Many other papers like PETR directly regress pose with the decoder. Then it is natural to ask Why use normalizing flow? What is the impact if we directly regress the pose? These two points may need further discussion.

3. What is the exact text prompt used in the given method?

4. The authors also claimed that the previous method ignored the global spatial relation between points. Yet it is still unclear to me how the author's method solves this problem. In the introduction, it is claimed that transformers are used to solve the problem. But it is too general to convince me.

**Limitations:**

No negative societal impact are found.

---

> ### Author Rebuttal · Authors · 2024-08-07
>
> W1. **Qualitative Results:** More qualitative comparisons with ViTPose and HRNet are shown in Figure 1 of the rebuttal document. From the figure, it is evident that excluding the stick from the bounding box is much more beneficial as it reduces the noise present in the image.
>
> ---
>
>
> W2. **t-SNE visualization of text embeddings:** Figure 1(b) of the main paper depicts the t-SNE visualization of the text embeddings generated from keypoint-specific text prompts. The two axes x and y represent the two reduced dimensions dimension1 and dimension 2 respectively. The different colours represent whether they are upper-body or lower-body joints, with red representing upper-body joints, blue representing lower-body joints and green representing the hip joints. In the paper, we mention that from the figure, it is evident that while these embeddings maintain positional structure within the upper-body joints (the shoulders are placed above the elbows which are placed above the wrists, following the pose of a human standing normally) and the lower-body joints (knees placed above ankles), they fail to maintain positional structure between the upper-body and lower-body joints (elbows and wrists are placed below knees and ankles in the plot which does not follow the pose of a human being). This has been explained in lines 41 to 46 in the introduction of the paper.
>
> ---
>
>
> W3. **Improvement on unseen keypoints:** As demonstrated in Table 1, the unseen keypoints (stick top, stick heel, and stick toe) contribute significantly to the overall improvement of TokenCLIPose. Specifically, an  improvement of 4%, 6.4%, and 7.35% is observed on the three keypoints respectively when compared to the second-best model (ViTPose). On the other hand, the seen keypoints (human) only improve by 1.76\%. Similar trends are observed when experimentation was done on the Lacrosse dataset in Table 2. Hence, it is evident that major improvements come from the unseen keypoints.
>
> ---
>
> Q1. We intended to say that most of the state-of-the-art works [1-4] follow a top-down approach. However, we will revisit this statement and update it in our camera-ready version.
>
> ---
>
>
> Q2. Thank you for noting our use of normalizing flows and asking an insightful question! We would like to explain this choice of ours by first stating the limitations of standard regression losses and then explain the reason behind using normalizing flows.
> **Limitations of standard loss functions:** Directly regressing poses using a standard $\ell_2$ or $\ell_1$  loss is not desirable as it is vulnerable to ambiguous and noisy labels [5] which is prevalent  in our setting due to high motion-blur. This is primarily because applying a standard distance function as our loss function is equivalent to assuming the distribution of the output probability distribution. For example, the loss function when we assume the output probability distribution to be a Gaussian with constant variance is equivalent to the standard $\ell_2$ loss. Similarly, the loss function when we assume the output probability distribution to be a Laplacean with constant variance is equivalent to the standard $\ell_1$ loss.
>
> **Why normalizing flows?** On the other hand, we learn the deviation between the predicted probability distribution and ground-truth probability distribution, which is estimated by a flow model, and minimize the deviation using MLE. By doing so, we get more robust predictions. We validate the impact of this Residual log likelihood (RLE) loss through an ablation study by replacing it with the standard $\ell_2$ loss. Upon doing so, we get a mean PCKh accuracy of 75.22\%, resulting in a 1.06\% performance drop.
>
> ---
>
>
> Q3. Addressed in C2 of the author rebuttal
>
> ---
>
>
> Q4. **Our difference over existing multimodal methods:** Existing multimodal 2D pose estimators [6] align the image features around each keypoint to their corresponding text embeddings using a contrastive loss. This pushes the image features to be similar to the text embeddings. However, this is not desirable as the text embeddings do not capture global positional structure (showcased in figure 1(b) of the main paper and discussed in W2). This is what we explain in lines 41-46 of the main paper. In order to overcome these limitations, we use these text embeddings as priors and initialize our learnable keypoint tokens (also referred to as text tokens) using these text embeddings. This provides our model with the inductive priors required for robust prediction without biasing them completely towards the text embeddings. This is mentioned in lines 47-51 in the main paper.
>
> ---
>
> References:-
>
> [1]: Xu et al., ViTPose: Simple Vision Transformer Baselines for Human Pose Estimation, NeurIPS 2022.
>
> [2]: Sun et al., Deep High-Resolution Representation Learning for Visual Recognition, CVPR 2019.
>
> [3]: Li et al., TokenPose: Learning Keypoint Tokens for Human Pose Estimation, ICCV 2021.
>
> [4]: Yang et al., TransPose: Keypoint Localization via Transformer, ICCV 2021.
>
> [5]: Li et al., Human pose regression with residual log-likelihood estimation, ICCV 2021.
>
> [6]: Hu et al., LAMP: Leveraging Language Prompts for Multi-person Pose Estimation, IROS 2023.

---

> > ### Comment · Reviewer_9aCe · 2024-08-11
> >
> > Thank you for your detailed response. Most of my concerns have been properly addressed, though I have a few additional points I’d like to discuss:
> >
> > 1.  As part of the ongoing discussion regarding the use of normalizing flow, I believe the method provides an explicit probability distribution for keypoint estimation, correct? If so, I’m curious whether the standard deviation of each prediction can be directly used as an uncertainty measurement or confidence score. If the method already employs this approach, could you please share the performance results? If not, I suggest testing this on the CrowdPose dataset, as its evaluation protocol (based on COCO) is highly sensitive to accurate confidence estimation of poses. Demonstrating improved performance in this area could potentially strengthen the paper.
> >
> > 2. Several reviewers have noted that the test dataset appears limited and relatively simple in terms of occlusion challenges. I recommend evaluating the method on the OCHuman [1], which has gained popularity recently for methods addressing occlusion. This could provide more robust evidence of the method’s effectiveness under challenging conditions.
> >
> > 3. Regarding the concern (C3) on how the text guides the model in predicting the exact location of unseen points, it would be helpful if the authors could visualize the attention map between the text prompt and the visual features (e.g., between F_text and F_vis or between F_text and F_loc). My assumption is that the attention between F_vis and F_loc (both visual features) struggles to focus on out-of-box regions because these regions (e.g., white regions if I am correct) lack visual similarity to in-box visual features. Thus, there is no hope to recover any useful information through the these visual features (F_vis and F_loc). However, text features might not rely on visual similarity and could direct attention to broader regions, including out-of-box areas.  In my point of view, this visualization could be more persuasive to other reviewers than simply citing other papers to support the intuition behind this idea.
> >
> > I understand that addressing these points may require additional experiments, which would take time. Therefore, I won’t factor the response to these discussions into my final rating, even if no further experimental results are provided.
> >
> > Finally, I noticed that most reviewers initially gave a negative rating, and I agree with many of their proper and valid points. I would like to see their feedback on the rebuttal before making my final decision. Also hope it will help to convince other reviewers through address a few of my concerns.
> >
> > [1] Pose2Seg: Detection Free Human Instance Segmentation

---

> > > ### Author Response · Authors · 2024-08-11
> > >
> > > We would like to thank the reviewer for their invaluable comments and suggestions.
> > >
> > > 1. **Standard deviation as confidence score**: Thanks for noting such minute details and raising this point! The results that we reported are using the standard deviation directly as a confidence measure.
> > >
> > > 2. **OCHuman Evaluation**: We appreciate the reviewer’s suggestion of evaluating our model on OCHuman. However, since the OCHuman dataset does not contain any train set, the general scheme followed by the research community is to train the model on COCO dataset and test it on OCHuman. This is difficult for us due to memory constraints as one complete training run on the COCO dataset takes approximately 4-5 days. Hence, this would not be possible before the rebuttal deadline. Moreover, in order to showcase robustness under occlusions, we did conduct experiments on CrowdPose, the largest occlusion benchmark.
> > >
> > > 3. **Impact of text prompts**: Thanks for the suggestion. We would like to point out that apart from citing other relevant research, we have also conducted quantitative experiments (Tables 4 and 6 of the main paper) showcasing the benefit of using text prompts on the accuracy of our model. Hence, we believe we have successfully demonstrated the positive impact of text prompts. Moreover, we agree that adding visualisations will further bolster our motivation behind utilizing text and will include them in our camera-ready version.

---

### Official Review · Reviewer_AetR · 2024-07-03

**Soundness:** 3
**Presentation:** 3
**Contribution:** 3
**Rating:** 4
**Confidence:** 5

**Summary:**

This paper proposes a text-guided keypoint localization method that can detect keypoint that is out-of-input image. The proposed method only crops person area and abandons the object area, then adopts language prior to predict keypoint that is out-of-input image. To verify the effectiveness of the proposed method, this paper also introduces a Ice-Hockey keypoint detection dataset. Experiments on CrowdPose demonstrate the effectiveness of the proposed method.

**Strengths:**

1.	Utilizing text to represent the spatial relationships of keypoints is reasonable and straightforward. The proposed TokenCLIPose can effectively model the keypoint relationship and estimate unseen keypoints accurately.
2.	This paper is well-written and easy to understand.

**Weaknesses:**

The proposed method is not well suited to the introduced tasks such as  Ice-Hockey. Abandoning the visible object is not a good solution because the model cannot make perfect prediction on out-side objects. For occluded keypoints, it is not bad because we can only guess the position, it is acceptable that the prediction is not so accurate. But for visible keypoints of hockey stick in Fig. 1(a), the predicted results are not acceptable. A better way to handle unnecessary visual features is to separate the object and person and perform keypoint localization on each individually.

I think the proposed method can be used to perform the occluded/out-of-box keypoint localization, but not to solve the Human-Object Keypoint Localization tasks such as Ice-Hockey. The introduced task and dataset cannot fully demonstrate the effectiveness of the proposed method. It is better to introduce the method from the perspective of occluded or out-of-box keypoint localization, abandoning the Ice-Hockey tasks and background and conduct more experiments on occluded  keypoint localization benchmarks such as OCHuman.

**Questions:**

In Weakness

**Limitations:**

The authors say they show the failure cases of the prooposed method, but I do not find them in the manuscript.

---

> ### Author Rebuttal · Authors · 2024-08-07
>
> W1. (a) **Excluding the sticks:** The concern of abandoning the visible object is addressed in the author's rebuttal.
>
> (b) **Individual localization of human and stick:** We would like to thank the reviewer for raising an insightful question of estimating the pose of human and their extension separately. While this might seem interesting, works on human-object interaction [1-3] and 2D pose estimation [4] showcase that the pose of the human and the extension are closely related, and can be used to refine each other’s pose. Our experiment on individual and joint pose estimation shown in Table 4 of the rebuttal document also demonstrates an improvement in the accuracy of human pose when jointly trained with the stick.
>
> (c) **Efficacy of our method to solve human-object Keypoint Localization:** We also appreciate the reviewer's suggestion of introducing this paper from the perspective of occluded keypoint localization. However, we disagree with the reviewer’s comment that our method cannot solve human-object keypoint localization. Through our extensive qualitative and quantitative experiments, we have demonstrated the superior performance of our model in comparison to SOTA approaches.
>
> ---
>
> References:-
>
> [1]: Yao et al., Modeling mutual context of object and human pose in human-object interaction activities, CVPR 2010.
>
> [2]: Gupta et al., Observing Human-Object Interactions: Using Spatial and Functional Compatibility for Recognition, TPAMI 2009.
>
> [3]: Delaitre et al., Learning person-object interactions for action recognition in still images, NeurIPS 2011.
>
> [4]: Neher et al., HyperStackNet: A Hyper Stacked Hourglass Deep Convolutional Neural Network Architecture for Joint Player and Stick Pose Estimation in Hockey, CRV 2018

---

### Official Review · Reviewer_F7X8 · 2024-07-06

**Soundness:** 2
**Presentation:** 2
**Contribution:** 2
**Rating:** 4
**Confidence:** 4

**Summary:**

The paper argues that using images including both humans and interacting objects introduces unnecessary visual features that hinder pose estimation performance. To address this, the paper claims that treating objects as unseen to predict interacting object poses can achieve better results and proposes a TokenCLIPose solution. The method is evaluated on the Hockey and Lacrosse datasets, which contain images from a total of 11 video clips.

**Strengths:**

The paper is generally clear and the method uses language information to improve pose estimation.

**Weaknesses:**

The experiments and comparisons are not convincing and the experiments are weak.

(1) The paper conducts its main experiments on the Hockey and Lacrosse datasets. However, these datasets are too small, comprising only 11 video clips.

(2) The paper claims that predicting object poses using human-only images performs better than using human-and-object images. However, the paper lacks in-depth analysis, and the experiments are insufficient to support this claim. The reviewer disagrees, arguing that observation is more accurate than imagination when an appropriate framework is used.

(3) The major contribution of the paper is not clearly defined.

(4) The paper did not discuss and compare with bottom-up methods, which are popular for human pose estimation.

(5) In Line 136, the text encoder simply encodes class-related information/names of each keypoint, which the reviewer argues cannot tackle the challenge of unseen keypoints in the image.

(6) Typos, such as "we curate two new sports dataset" and "Experimentations".

**Questions:**

See Weaknesses.

**Limitations:**

No discussion on limitations is provided.

---

> ### Author Rebuttal · Authors · 2024-08-07
>
> W1. (a) **Custom dataset size:** To the best of our knowledge, this is the first work to address the problem of extension pose estimation and create a dataset for the task. Since pose estimation is a task that is heavily dependent on labels, we spent a lot of time collecting precise manual annotations. Though the reviewer might feel 11 video clips might be less, the number of annotations that we collected still sum up to a sizable 11630 images for ice hockey and around 300 for Lacrosse (used for zero-shot pose estimation).
>
> (b) SOTA results on large dataset (CrowdPose): Moreover, one of our main experiments is conducted on the CrowdPose dataset (Table 3 of the main paper) is a benchmark dataset from the literature that has atleast four times more human images when compared to the Ice hockey dataset.  The results show that TokenCLIPose outperforms existing state-of-the-art architectures on this CrowdPose dataset as well. This shows that irrespective of the dataset size, our network performs better when keypoints are unseen.
>
> ---
>
> W2. **Exclusion of stick from the bounding box:** This is addressed in C1 of the author rebuttal.
>
> ---
>
> W3. **Main contributions:** Our main contributions are as follows:-
> - We reformulate the extension pose estimation problem as an out-of-bounding box keypoint detection problem by explicitly leaving out the extensions, and leverage the human features and keypoint-specific text prompts to robustly estimate the human and extension pose.
> - Existing multimodal pose estimators try to align the image features with the keypoint-specific text embeddings using a contrastive loss. However, upon visualizing these text embeddings, we find out that this is not desirable as the embeddings do not maintain global structure (as shown in Fig. 1(b) of the main paper). Hence, we use them as additional priors and initialize our learnable keypoint tokens (also referred to as text tokens) with these text embeddings rather than biasing our model completely towards these text embeddings.
> - Realizing the lack of datasets for extension pose estimation, we introduce 2 datasets: an ice hockey and a Lacrosse dataset. Furthermore, we show that our model outperforms the existing SOTA approaches by 4.36\% and 2.35\% respectively.
> - We further demonstrate our model’s robustness to occlusions by conducting experiments on the CrowdPose dataset, where we outperform the SOTA top-down approaches by 3.8\%.
>
> ---
>
> W4. **Why not bottom-up?** Most of the SOTA techniques [2-5] implicitly follow a top-down approach. This is mainly because they lead to more accurate predictions [1]. Hence, we compare with top-down approaches.
>
> ---
>
> W5. **How do text prompts contain exact location?** This is addressed in C3 of the author rebuttal.
>
> ---
>
>
> W6. Thanks for spotting the typos. We will update it for the camera-ready submission.
>
> ---
>
> References:-
>
> [1]: Jiang et al., RTMW: Real-Time Multi-Person 2D and 3D Whole-body Pose Estimation, arXiv 2024.
>
> [2]: Xu et al., ViTPose: Simple Vision Transformer Baselines for Human Pose Estimation, NeurIPS 2022.
>
> [3]: Sun et al., Deep High-Resolution Representation Learning for Visual Recognition, CVPR 2019.
>
> [4]: Li et al., TokenPose: Learning Keypoint Tokens for Human Pose Estimation, ICCV 2021.
>
> [5]: Yang et al., TransPose: Keypoint Localization via Transformer, ICCV 2021.

---

> > ### Comment · Reviewer_F7X8 · 2024-08-14
> >
> > Thank you for the authors' responses. However, like Reviewer 9Npr, many of my concerns remain unaddressed. The experiments do not convincingly support the main claim.

---

### Official Review · Reviewer_9Npr · 2024-07-09

**Soundness:** 2
**Presentation:** 2
**Contribution:** 2
**Rating:** 4
**Confidence:** 3

**Summary:**

This paper introduces the problem of estimating the keypoints of humans together with a stick-like object that the human interacts with, i.e. hockey stick or lacrosse stick (which is referred to as "extension" in this paper). The paper claims that prior works on 2D human pose estimation cannot be naively extended to this task by simply using a larger cropped image containing the human and the object together, because it would introduce unnecessary features and hurt performance. To resolve this issue, the paper proposes the TokenCLIPose method, which focuses solely on human using a tight crop bounding box and utilizes keypoint-specific text embeddings from CLIP to predict out-of-bounding-box keypoints of the stick objects. Specifically, it uses a CNN to extract features from images and obtain a coarse estimation of human keypoints in the bounding box with an MLP. Subsequently, it feeds the image features, coarse keypoints features and the keypoint-specific text embeddings of the stick keypoints from a CLIP text encoder to a transformer to obtain a final keypoint estimation result. The experimental results indicate their method outperforms previous 2D human keypoint estimation methods in their newly curated datasets for this new setting, as well as on the CrowdPose dataset with multi person scenarios that have severe occlusion.

**Strengths:**

1. The paper is reasonably well-structured and the motivation and approach are described well.
2. The integration of features from foundation models, like CLIP, for keypoint detection is original in general, but I am unsure why it would be beneficial in the particular application scenario proposed in this paper.
3. The experiments on various dataset show that the proposed method outperforms related works.

**Weaknesses:**

1. The proposed problem of detecting keypoint of cropped out objects seems artificial, and it remains unclear if predicting the keypoints of cropped objects is actually necessary. One important ablation study that needs to be shown is to train TokenCLIPose on images that are not artificially cropped, and instead contain both the human and the stick together. It seems that if the text embedding is helpful for locating keypoints of unseen objects, it should also help when the in the bounding box.
2. The description of adopting the keypoint-specific text embeddings for predicting the keypoints of unseen keypoints is unclear. Why should the text embeddings contain any information of the location of unseen objects in not explained and not studied sufficiently in the experiments.
3. How are the baseline methods in the experiments adapted to the newly proposed task? This is not discussed at all. The number of keypoints for the extensions seems to vary between experiments. How do you adapt human pose estimation methods for this new task and why do these methods fail in this setting? This should be explained in more detail.
4. The proposed method should be compared to more recent works like ViTPose++ (TPAMI 2023) and BUCTD (ICCV 2023). The latest works used for comparison in the submission are ViTPose (NeurIPS 2022) and HRFormer (NeurIPS 2021). As this paper works on a different setting than classical human pose estimation, it is important to discuss why existing 2D human pose estimation methods are limited in dealing with this setting.
5. What are the text prompts used for the Text-based Keypoint Encoder? There is no explanation regarding this essential part of the method.
6. This method is only verified with simple objects like hockey sticks and lacrosse sticks. A core open question is how the method would generalize to more complicated structures like chairs?

**Questions:**

In addition to the questions contained in the previous section:
1. How is the visualization of Figure 1(b) obtained?
2. For the ablation study in Table 4, what is the result of only combining text token and image token?

**Limitations:**

The paper shows failure cases, but only limited quantitative results are shown and there is no discussion about these failure cases.

---

> ### Author Rebuttal · Authors · 2024-08-07
>
> W1. **Training TokenCLIPose with sticks included:** To assess TokenCLIPose's performance on uncropped images, we extended our experiments on the uncropped images (images that contain both players and their sticks). As shown in Table 2 of the rebuttal document, the performance drops by 4.51\% resulting in a mean accuracy of 71.77\%. These results reinforce our hypothesis that expanding the field-of-view introduces unnecessary visual features which hinder model accuracy.
>
> ---
>
> W2. **Impact of text prompts:** This is addressed in C3 of the author rebuttal.
>
> ---
>
> W3. (a) **Baseline training details:** Baseline methods are trained from scratch on the extended bounding boxes (boxes that contain human and stick). All models are trained on the Ice hockey dataset to predict 15 keypoints (12 human + 3 stick). For the zero-shot setting, we directly use the ice hockey trained models on our Lacrosse dataset and infer predictions. Specifically, we discard the 15th keypoint predicted by the model, since the Lacrosse dataset has only 14 points (12 human + 2 stick). The human keypoints are the same for both the Lacrosse and Ice hockey datasets.  We understand that we have failed to include this detail and will update this in the camera-ready version.
>
> (b) **Why do Baselines fail?** These methods fail in this setting primarily due to the expansion of the bounding boxes, which increases the field-of-view. This introduces unnecessary visual features as depicted in Figure 3 of the rebuttal document. which confuses the model and leads to suboptimal predictions.
>
> ---
>
> W4. (a) **Why not ViTPose++ and BUCTD?** ViTPose++  deals with heterogeneous body keypoint categories thereby adopting task-agnostic and task-specific feed-forward networks in transformers. BUCTD uses a hybrid approach that combines the strengths of bottom-up and top-down methods. Thus, since we focused on improving over top-down approaches with task-specific fixed keypoints we avoided both these frameworks.
>
> (b) **ViTPose++ results:** However, for completeness, we trained ViTPose++ on the ice hockey dataset which resulted in a mean PCKh of  72.09\% underperforming by 4.19\% when compared to TokenCLIPose as shown in Table 3 of the rebuttal document.
>
> ---
>
> W5. **What are the text prompts?** This has been explained in C2 of the author rebuttal.
>
> ---
>
> W6. **Experiments on “simpler” objects:** To the best of our knowledge, this is the first work to address the problem of extension pose estimation and create a dataset for the task. Since pose estimation is a task that is heavily dependent on labels, we spent a lot of time collecting precise manual annotations. Also, the setting in which the objects are considered is highly dynamic, making the task extremely challenging. This can be seen in Tables 1 and 2 of the main paper as existing SOTA methods struggle to estimate the object pose. Hence, we politely disagree with the reviewer that we consider simple objects and find it unfair to criticize us on the data front as we are the first to address the problem.
>
> ---
>
> Q1. **Figure 1(b) explanation:** The visualization in Figure 1(b) of the main paper is a t-SNE plot of each of the keypoint prompts used to describe the human keypoints.
>
> ---
>
> Q2. **Text and image tokens ablation:** Combining text and image tokens resulted in a mean PCKh of  75.72\%. Thank you for spotting the missed ablation. We will add it in the camera-ready version.

---

> > ### Comment · Reviewer_9Npr · 2024-08-12
> >
> > I appreciate the effort of the authors, however, it remains unclear to me why cropping out the object of interest from the image actually improves the performance. The argument made is that this introduces irrelevant features, but this should only be a problem in a limited data setting where the model starts overfitting on the background, is that correct? This hypothesis could be even verified with synthetic data rendered with computer graphics in cluttered backgrounds. To me, and it seems also several other reviewers, the paper does not sufficiently study when and why the model fails when the stick is included in the crop. Demonstrating a performance decrease in the results is a necessary initial step, but since the result is very unintuitive, I would expect a more in-depth study of the reasons for this problem.

---

> > > ### Author Response · Authors · 2024-08-13
> > >
> > > Thank you for taking the time to review our rebuttals. We believe we have conducted comprehensive experiments that sufficiently illustrate the rationale behind our approach of cropping out sticks. Specifically:
> > >
> > > 1. **Noise Influence:** We have demonstrated the influence of noise by comparing our model with ViTPose on two different data splits—one with predominantly visible keypoints and another with a mix of visible and noisy keypoints, as presented in Table 1 of our rebuttal document.
> > >
> > > 2. **Training on Human+Stick Bounding Boxes:** We have further validated our approach by training our model on datasets containing human+stick bounding boxes, as shown in Table 2 of our rebuttal document, which underscores the efficacy of our work.
> > >
> > > 3. **Qualitative Performance:** Finally, we have provided qualitative evidence of our model's superior performance in noisy conditions in Figure 1, where noisy regions are clearly marked.
> > >
> > > Given these extensive evaluations, we are unsure what additional experiments would be necessary to further validate our hypothesis. If there are specific experiments or analyses you would like to see, we would greatly appreciate your guidance in this matter.

---

> > > > ### Comment · Reviewer_9Npr · 2024-08-13
> > > >
> > > > Unfortunately, my questions and comments from the previous post were not addressed. To me, the proposed task (predicting keypoints of sticks held by humans outside a cropped bounding box) is not sufficiently well motivated (which is also stated by reviewers F7X8 and AetR). I agree with reviewer F7X8 that the problem is very likely caused by the small amount of training data (in total only 11 videos, which is very few data in the context of keypoint detection). Hence, the overall relevance of this work is reduced to detecting keypoints of sticks held by humans from a small set of videos, which seems a rather niche area of research. While the proposed method does lead to some improvements in performance, I am not convinced of the general relevance of the contribution, which would need to be shown on a larger variety of object classes, not just sticks.

---

> > > > > ### Author Response · Authors · 2024-08-13
> > > > >
> > > > > **Dataset Size**: As mentioned in W1(a) of our response to reviewer F7X8, to the best of our knowledge, this work is the first to address human extensions and their pose estimation. Given this, there is no public dataset available for this specific task. Pose estimation requires precise annotations, which cannot be effectively generated by large language models (LLMs) or vision-language models (VLMs). Therefore, manual annotation was necessary, and this process is extremely time-consuming.
> > > > >
> > > > > We believe it is important to recognize the significant effort required to manually create this dataset for the first time. As such, it would be unjust to criticize our dataset size given these constraints. Nonetheless, we really appreciate your comments and will try to collect more data to showcase the impact of our approach.

---

### Author Rebuttal · Authors · 2024-08-07

Dear Reviewers,

We would like to thank all the reviewers for providing constructive feedback that helped us improve the paper. We are delighted that the reviewers recognized the originality of our work in incorporating language to predict out-of-bounding box keypoints (R1, R4), significance of our focus on the interesting topic (R4), the effectiveness of our method in modeling spatial relationship between keypoints (R3), and the substantial improvement over the existing SOTA (R1, R4). Throughout the rebuttal, we refer to the reviewers in the following manner: {Reviewer 9Npr: R1, Reviewer F7X8: R2, Reviewer AetR: R3, Reviewer 9aCe: R4}.

We address some of the common concerns here. The concerns have been paraphrased as multiple reviewers had similar concerns and questions.

---

### C1. Counter-intuitiveness of excluding the extension from the bounding box (R2 and R3) ###
**Training TokenCLIPose with sticks included:** To showcase the effectiveness of cropping out the hockey stick, we train our model on the extended bounding boxes (containing both the human and stick) and report the results in Table 2 of the rebuttal document. From the table, we can see that including the stick in the bounding box deteriorates our model’s performance by 4.51\%, resulting in 71.77\% mean accuracy.

**Impact of noise:** Furthermore, in order to investigate the impact of noise on the final predictions, we record the performance of ViTPose and TokenCLIPose under two different data split-ups in Table 1 of the rebuttal document. Specifically,  the ice hockey dataset is divided into 2 sets, one with images that predominantly contain visible keypoints (referred to as A) and the other with images that contain both visible and noisy (partially visible) keypoints (referred to as B). The results demonstrate that ViTPose outperforms TokenCLIPose (ours) by 2.49\% when only split-up A was used but underperforms by 4.36\% when split-up B is used. While our method might seem counter-intuitive, these experiments explicitly demonstrate that existing models aren't robust enough to noise necessitating the need for smaller bounding boxes. The distribution of the visibility of keypoints across frames is shown in Figure 2 of the rebuttal document.

---

### C2. What are the text prompts for each keypoint? (R1 and R4) ###
The text prompts corresponding to each keypoint used in TokenCLIPose are as follows:-

| Joint          | Text Prompt                                                                       |
|----------------|-----------------------------------------------------------------------------------|
| Left Shoulder  | left shoulder of a person                                                         |
| Right Shoulder | right shoulder of a person                                                        |
| Left elbow     | left elbow of a person                                                            |
| Right elbow    | right elbow of a person                                                           |
| Left wrist     | left wrist of a person                                                            |
| Right wrist    | right wrist of a person                                                           |
| Left hip       | left hip of a person                                                              |
| Right hip      | right hip of a person                                                             |
| Left knee      | left knee of a person                                                             |
| Right knee     | right knee of a person                                                            |
| Left ankle     | left ankle of a person                                                            |
| Right ankle    | right ankle of a person                                                           |
| Stick Top      | top portion of the stick opposite to the blade of a hockey stick held by a person |
| Stick Heel     | intersection of the blade and the shaft of a hockey stick                         |
| Stick Toe      | end portion of the blade that is farthest away from the shaft of a hockey stick   |

---

### C3. How do text embeddings contain any information about the exact location of the unseen keypoints? (R1 and R2) ###
Our choice of using text prompts to tackle the challenge of unseen keypoints is guided by recent works on 3D pose estimation [1] which use language to interpolate missing poses temporally, and image inpainting [2-3]. Furthermore, the authors have demonstrated in Tables 4 and 6 of the main paper that the proposed network drops in performance significantly without text prompts (by 3.35\%).

---

**Summary:** We address a critical yet overlooked problem: joint pose estimation of humans and extensions (objects that humans hold and interact with), and overcome the limitations of existing SOTA top-down approaches on this task. Our approach is counter-intuitive yet impactful: we propose excluding the extension from the bounding box and predicting keypoints beyond the bounding box, thereby challenging the standard practice of including all keypoints within a bounding box. To achieve this, we utilize language priors along with bounding box features through keypoint-specific text prompts. To evaluate our method, we introduce the first dataset with labeled human and extension poses from broadcast ice hockey videos. We also perform zero-shot evaluations on a self-curated Lacrosse dataset to showcase our model’s generalizability. Finally, to showcase our model’s robustness to occlusions, we conduct experiments on the CrowdPose dataset.

Please see our reviewer-specific feedback for more information.

References:-

[1]:  Tevet et al., Human Motion Diffusion Model, ICLR 2023.

[2]:  Ni et al., NUWA-LIP: Language-guided Image Inpainting with Defect-free VQGAN, CVPR 2023.

[3]: Zhang et al., Text-Guided Image Inpainting, MM 2020.

---

### Decision · Program_Chairs · 2024-09-25

**Decision:**

Accept (poster)

**Comment:**

The reviewers agree that the proposed method is clear and that the experiments show reasonable improvement over the state of the art.  There were some questions about whether the range of datasets evaluated was sufficient, but after the rebuttal and discussion, the reviewer closest to the area and the area chair both found this sufficient.  The reviewers make some comments about explaining the context and advantage of the proposed method that should, if possible, be utilized in cleaning up the paper.